# Meltblow Processing of Poly (Ethylene Furanoate)–Bio-Based Polyester Nonwovens

**DOI:** 10.3390/ma18030544

**Published:** 2025-01-24

**Authors:** Tim Hiller, Hagen J. Altmann, Iris Elser, Mehdi Azimian, Michael R. Buchmeiser

**Affiliations:** 1German Institutes of Textile and Fiber Research Denkendorf, Koerschtal Str. 26, 73770 Denkendorf, Germany; 2Institute for Polymer Chemistry, University of Stuttgart, Pfaffenwaldring 55, 70569 Stuttgart, Germany

**Keywords:** polymer, polyester, biopolymer, polycondensation, 2,5-furan dicarboxylic acid (FDCA), polyethylene furanoate (PEF), melt processing, rheology, nonwovens, meltblow

## Abstract

Poly(ethylene furanoate) (PEF) has been identified as a bio-based alternative or supplement to poly(ethylene terephthalate) (PET) for various applications such as food packaging and bottles as well as technical- and high-performance fibers and yarns. In this study, the processing of PEF nonwovens in the meltblow process is successfully demonstrated and reported for the first time, according to our best knowledge The resulting fabrics achieved median fiber diameters of 2.04 µm, comparable to PET. The filtration efficiency of 25 g m^−2^ fabrics exceeded 50% comparable to PET and PBT of the same grammage and was raised to over 90% with post-process electrostatic charging, maintaining stability. As for other aromatic polymers, applying infrared heating modules into the process indicated the potential to minimize heat shrinkage. However, the suppressed ring rotation and slower crystallization kinetics of PEF showed the need for longer post-treatment times as the heat shrinkage remained between 20% and 40% at 10 °C. Overcoming this, PEF can be a viable, bio-based alternative to PET, particularly for such high-temperature nonwoven applications that require thin layers.

## 1. Introduction

The demand for bio-based alternatives to oil-based polymers has increased. Poly(ethylene furanoate) (PEF) is a bio-based polymer with the potential to replace or substitute poly(ethylene terephthalate) PET due to its similarity in structure and properties. Both polymers are synthesized through melt polycondensation of ethandiol (monoethylene glycol, MEG) with a dicarboxylic acid (optionally followed by solid-state polycondensation (SSP) to further increase the molar mass) and can be produced on the same reactor platforms with minor adjustments of process parameters (temperature, pressure, and catalysts) without any machine-side modifications [1,2,3,4]. 2,5-Furan dicarboxylic acid (FDCA), the second monomer of PEF, is chemically similar to terephthalic acid (PTA), the second building block of PET. Substitution of the phenylene ring by a furan ring increases the polarizability and interaction between the polymer chains. This leads to superior thermal and mechanical properties (modulus and tenacity) [5,6] and significantly better barrier properties against gas, water, and carbon dioxide compared to PET [7,8]. In contrast to PET, the hindered rotation of PEF chains due to a more rigid backbone [6] leads to a decreased aromaticity and orientation [9,10]. If bio-based MEG is applied in the PEF synthesis, a 100% bio-based polymer is formed, aligning with one of the key goals of Germany’s “2030 National Research Strategy for BioEconomy”: “Industrial usage of renewable resources” [11]. Production chains for FDCA and its precursor 5-hydroxymethylfurfural (5-HMF) from waste biomass, such as chemo-enzymatical conversion from the roots of chicory plants, have already been demonstrated and lie outside the food chain [12,13,14,15,16]. This meets a further important aspect of the BioEconomy: “*[…] neither additional agricultural land is required, nor crude oil is consumed*” [17,18]. Another advantage of PEF over PET is the higher glass transition temperature (*T_g_*) of PEF (85–89 °C), which is almost 15 °C higher than the one of PET (70–76 °C) [6,19,20]. Also, the melting range of PEF (~210–215 °C) is about 40 °K lower than the one of PET (250–260 °C) [2,6,10,19,21]. This allows PEF to be used at higher service temperatures, while melt processing consumes less energy [1]. However, the high costs of FDCA, and thus of PEF (>>100 € kg^–1^ in 2023), have prevented a large-scale market entry. Prices are expected to drop with market expansion as previously observed for polylactic acid (PLA) [22].

To date, no studies on meltblow processing of PEF have been published. Several factors may contribute to this absence. First, the technology required to minimize heat shrinkage is patented [23,24,25], making it either exclusive or costly to access. Additionally, the high monomer costs of FDCA and the limited availability of PEF in sufficient quantities pose an obstacle, as industrial-scale meltblow lines require at least 50–100 kg of PEF per trial. Additionally, the quantitative importance of PET in the meltblown sector is significantly lower than that of polypropylene (PP). This is primarily due to PET’s slower crystallization kinetics [26,27], which are disadvantageous under the substantial initial temperature drop (up to 70 K) that occurs after the fibers leave the meltblow spinneret [28]. Common to all fiber-generating processes, the fibers produced via meltblowing are highly oriented. However, without stabilization of the polymer chains through crystal formation, they tend to revert to form entanglements when heated above *T*_g_. The resulting fabrics are low in strength and show high heat shrinkage. Using conventional meltblow lines, this issue is addressed with an additional process step or thermal post-treatment, such as calendering [29]. Consequently, poly(butylene terephthalate) (PBT) instead of PET is used as primary polyester for meltblown nonwovens despite its higher cost and lower long-term temperature stability (max. 150 °C) [30], due to its faster crystallization rate. Nevertheless, the material price of PET falls within the low-price segment of thermoplastic polymers. Furthermore, PET has partially superior properties compared to other polymers in this price range, with an application temperature of up to 200 °C, making it suitable for high-temperature filtration applications [31,32]. Polymers with comparable or higher temperature stability, such as PPS or PEEK, are currently more expensive (by a factor of 20–100) and more challenging to process or tailor for meltblow processing, requiring highly flowable, low-viscosity grades [28]. Previous developments have successfully addressed the shrinkage and web strength limitations of PET meltblown fabrics by precisely controlling the material temperature throughout the process—from the point at which the polymer melt exits the spinneret with the air stream, ideally up to the fabric winding [23,24,25,26,33] and over the fiber deposition area (conveyor belt). Consequently, the molten polymer and later the fibers are maintained above *T_g_* as long as possible [26,33], ensuring an optimal degree of crystallization and an elimination of heat shrinkage [26].

This study aims to use PEF in the meltblow process using a technical-scale meltblow line, achieving industry-relevant per-hole throughputs. Different PEF grades, characterized by varying intrinsic viscosities ([η]), are employed to assess the necessity of solid-state post-crystallization for achieving a highly flowable melt suitable for meltblowing. Established PET and PBT grades are used as a benchmark for comparison against a commercial reference. Filtration efficiency, both with and without electrostatic charging, and the mechanical properties of the produced fabrics are evaluated. The degree of crystallinity state of the PEF and PET fibers after the meltblow process is analyzed using wide-angle X-Ray scattering (WAXS) and compared to both the virgin polymer and high-oriented melt-spun textile fibers.

## 2. Materials and Methods

### 2.1. Materials

#### 2.1.1. Chemicals

MEG was purchased from Brenntag GmbH (Essen, Germany). Titanium (IV) tetra(*n*-butoxide) (≥99%, Alfa Aesar, Ward Hill, MA, USA) was obtained from VWR International LLC. Triphenyl phosphate (p.a.) was purchased from Merck KGaA (Darmstadt, Germany). FDCA (>99%) was acquired from Purac Biochem (Gorinchem, The Netherlands) and used as received.

#### 2.1.2. Synthesis of PEF

The synthesis of PEF was achieved by esterification of FDCA and MEG (Figure 1), followed by melt polycondensation. The synthesis protocol is based on earlier publications [1].

Syntheses were accomplished on a 25 kg scale. For PEF synthesis, a 50 L steel autoclave (Juchheim GmbH, Bernkastel Kues, Germany with a pressure stability of up to 15 bar) was charged with FDCA (1 eq., 166.6 mol, 26.0 kg), MEG (2.2 eq., 366.5 mol, 22.8 kg), and titanium (IV) tetra(*n*-butoxide) (0.000225 eq., 0.0375 mol, 12.7 g) as well as triphenyl phosphate (0.000225 eq., 0.0375 mol, 12.2 g). The reactor was flushed with nitrogen and then sealed and heated to 200 °C for 4–7 h. Water was released stepwise via a column and a condenser. Once sufficient water was removed, the pressure was reduced in several steps below 1 mbar and a temperature of 260 °C was applied for several hours. The polymerization progress was monitored by analysis of the distillate (refractive index) and the torque of the stirrer. The polymer was discharged from the reactor in the form of a strand and solidified in an ice bath directly after the discharge valve. These strands were pelletized and dried under ambient air.

#### 2.1.3. Solid-State Polycondensation

The molar mass of the produced polymers was increased and adjusted by consecutive SSP. PEF, as synthesized, was treated in a tumble drier under vacuum. A sub-atmospheric pressure of <0.1 mbar was applied at a temperature of 195 °C. The PEF pellets were kept isothermal for 3 days.

#### 2.1.4. Commercial Benchmark Materials

PET “Advanite 64001” granule was purchased from Advansa GmbH (Hamm, Germany, with a [η]-value of 0.550 ± 0.020 dL g^–1^ (K070) and a (COOH-) end-group concentration of 30.0 eq. ton^–1^ (K069) [34].

Pocan B600 (TP010-002) was obtained from LANXESS Deutschland GmbH (Köln, Germany), with a melt volume-flow rate of 225 cm^3^ 10 min^–1^ at 250 °C, 2.16 kg (ISO 1133-1), and melting temperature of 225 °C (ISO 11357-1,-3). The density was 1.310 g cm^−3^ (ISO 1183) [35].

Both polymers were rheologically characterized in a previous study [28] for their suitability in the meltblow process.

#### 2.1.5. Crystallization and Drying

PEF was crystallized in a vacuum oven at 120 °C for 1 day prior to use. Before usage, all polymers were dried in an oven for >6 h at 80 °C under vacuum (<1.8 × 10^–1^ mbar).

### 2.2. Polymer Characterization

#### 2.2.1. Intrinsic Viscosity Measurements

The polymers were characterized on their solution viscosity in accordance with DIN EN ISO 1628-5 at 25 °C using an Ubbelohde-Ia viscometer in dichloroacetic acid (99%).

#### 2.2.2. Carboxyl End-Group Content

The content of carboxyl end groups (CEGs) of the PEF samples was determined by potentiometric titration in an m-cresol/dichloro-methane mixture according to ASTM D7409-15 with a KOH solution in isopropanol used as a standard solution. The average of two independent measurements was used for each sample.

#### 2.2.3. Differential Scanning Calorimetry (DSC)

DSC measurements were carried out under continuous nitrogen flow (20 mL min^−1^) on a Q2000 differential scanning calorimeter (TA Instruments Inc., New Castle, DE, USA) with a heating rate of 10 K min^–1^. The sample mass was 2 mg to 10 mg. The melt enthalpy Δ*H_m_*, melting peak temperature *T*_m,p_, and *T*_g_ were determined from the heat flow–temperature curves. As the (re−)crystallization of PEF is very slow and for this reason not displayable using common standard DSC settings [1]. Therefore, for both the granules and the fiber samples, only one measurement per sample was carried out, which was limited to the 1st heating cycle only.

The degree of crystallinity *χ_c_* was calculated by standardizing the melt enthalpy to the standard melt enthalpy Δ*H_m,0_*, as shown by Equation (1):(1)$$\chi_{c}=\frac{\Delta Hm}{\Delta Hm,0}$$

The standard melt enthalpy Δ*H_m_*_,0_ was 137 J g^−1^ for PEF [19], 145 J g^−1^ for PBT [36], and 140 J g^−1^ [37,38] for PET in accordance with the literature.

#### 2.2.4. Determination of the Moisture Content

The residual water content in the polymer samples was measured via Karl Fischer titration, which was carried out on an “899 Coulometer” and an “885 Compact Oven SC” (both: Deutsche METROHM GmbH & Co. KG, Filderstadt, Germany) at 140 °C. The water content was below 0.015 wt.% for all polymers.

#### 2.2.5. Rheological Characterization

Shear rheological experiments in the temperature and time-sweep modes were performed on a “Physica MCR 501” rheometer (Anton Paar Group AG, Graz, Austria) in plate–plate geometry. Polymer granules were placed on the lower plate (25 mm in diameter), and the gap was adjusted to 1.0 mm. Afterward, excess material was removed, and the test was performed under a nitrogen atmosphere (50 mL min^–1^). Temperature ramps (heat rate: 2 K min^–1^, strain: 10%, angular frequency: 10 rad s^–1^) were performed under adjustment of the gap in order to maintain a constant normal force over the measurement. The strain amplitude was adjusted to the linear viscoelastic regime by strain sweep tests at a constant angular frequency of 10 rad s^–1^. Measurements in the time-sweep mode (strain: 10%, angular frequency: 10 rad s^–1^) were conducted at selected process temperatures to prove the thermal stability of the materials.

#### 2.2.6. Wide-Angle X-Ray Scattering

Wide-angle X-Ray diffraction (WAXD) measurements were recorded on a D/Max Rapid II diffractometer (Rigaku Corp, Akishima, Japan) using monochromatic Cu *Kα* radiation (*λ* = 0.15406 nm; *U_acc_* = 40 V; *I_acc_* = 30 mA) and an image plate detector. A scanning rate of 0.2° min^−1^ and a step size of 0.1° were applied. The measurement time was 1 h for all investigated samples. The diffraction patterns were analyzed using the PDXL 2 software, and pseudo-Voigt profile fitting was chosen for the evaluation of reflex positions and crystalline fraction determination. The degree of crystallinity *χ_c_* was calculated according to Equation (2):(2)$$\chi_{c}=\frac{\sum I_{c}}{\sum \left(I_{c}+I_{a}\right)}$$
where *I_c_* and *I_a_* are the integrated intensities of crystalline reflexes and amorphous reflexes, respectively.

The samples were prepared by arranging nonwoven sheets parallel to the carrier.

#### 2.2.7. Size-Exclusion Chromatography (SEC)

For the determination of the molar mass distributions, an Agilent Technologies 1260 Infinity II High-Temperature GPC System (GPC 220, Agilent Technologies, Inc, Santa Clara, CA, USA) was used. It was equipped with a refractive index detector and operated at 50 °C in m-cresol as eluent. The polymer solution was prepared using 20 mg of the PEF samples, dissolved in 20 mL of m-cresol solution at 80–120 °C for 0.5–3 h. For the measurement, three consecutive PLgel Olexis columns (0.013 Å pore size) and one precolumn were used at a flow rate of 0.4 mL min^–1^. To record and analyze the chromatograms, the GPC/SEC software from Agilent Technologies (Santa Clara, CA, USA) was used. For calibration, narrowly distributed polystyrene standards with molar masses from 1681 to 2,000,000 g mol^−1^ were used.

### 2.3. Nonwoven Processing

#### 2.3.1. Meltblow Set-Up

Nonwoven processing tests were carried out on a technical-scale meltblow line of 500 mm working width. The system consists of a single-screw extruder (3-zone screw, ∅ 20 mm × 20 D) with a maximum throughput of 4 kg h^–1^ from Extrudex GmbH, Mühlacker, Germany), a gear pump from Mahr Metering Systems GmbH (Göttingen, Germany) with a volume of 0.6 cm^3^ rpm^–1^ a 561 and hole Exxon type spinneret with a width of 500 mm (28.4 holes per inch (hpi)). The nozzle capillaries show a diameter (D) of 0.3 mm with a length (L) to diameter ratio (L/D) of 8, determining a pressure limit of the spinneret of 50 bar. A safety limit was set at 45 bar. The temperature was applied along eight heating zones from the feed zone (zone 1) of the extruder to the spinneret (zones 6 to 8). The maximum die pressure of the spinneret was set at 50 bar with a safety limit of 45 bar. The air system consists of a compressor (Aertronic D12H) from Aerzener Maschinenfabrik GmbH (Aerzen, Germany) providing an air flow rate between Nm^3^ h^–1^ (minimum) and 325 Nm^3^ h^–1^ (maximum), combined with a flow heating system from Schniewindt GmbH & Co KG (Neuenrade, Germany). The setback between the nozzle tip and the air blades was 1.2 mm (non-variable), while the end gap could be adjusted for each polymer to achieve a homogeneous air/melt flow without melt adhesion to the air blades. The conveyor belt of Siebfabrik Arthur Maurer GmbH & Co KG (Mühlberg, Germany) was a steel mesh belt in canvas weave with clip seam in a total width of 0.72 m (no. 16 cm^–1^ linen weave) with a warp and weft wire (both 0.22 mm in diameter) made of stainless steel (1.4404 AISI 316L). The conveyor belt had a maximum winding speed of 10 m min^–1^ and could be adjusted in height relative to the spinning die from 200 mm to 500 mm in order to vary the die-collector-distance (*DCD*). Below the belt section on which the filaments were deposited, a suction box (suction area of 0.128 m^2^, 0.200 m × 0.640 m) with a maximum extraction volume of 2900 Nm^3^ h^–1^ (maximum flow rate: 11 m s^–1^) extracted the process (and secondary) air and supported the web formation on the belt.

#### 2.3.2. Nonwoven Production (Trials)

Meltblown nonwovens were produced from all polymers at varying process temperatures (according to the determined rheological properties, see Section 2.2.5) and polymer throughputs to determine the stable processing window. The melt temperature was adjusted over the temperature of the die and spinning head based on the results of the rheological characterization (see Section 2.2.5) of the respective material. The upper limiting and, thus, target value for the zero-shear viscosity at process temperature is defined as <150 Pa s^–1^. The temperature was further adjusted during the experiments in order to obtain a constant fiber formation at the die and a homogeneous shot-free laydown on the conveyor belt. The collector speed was adjusted to the polymer throughput to produce a constant area base weight of 25 g m^–2^ for comparability (without influence of the base weight) of web properties at different process settings. Furthermore, the *DCD* and the air temperature were optimized (stable fiber formation and homogeneous collection) and otherwise kept constant to minimize the experimental grid. The process air throughput was varied between minimal and maximal output (170–325 Nm^3^ h^–1^) of the compressor to define the possible diameter range for each polymer at the respective process setting.

To reduce undesirable heat shrinkage, a post-treatment with an infrared heater “MX 500/810” of Heraeus Noblelight GmbH (Kleinostheim, Germany) (“3” in Figure 2) between fiber deposition and winding was added to the process.

The infrared heater consisted of a metal case with a dimension of 0.50 m in MD (machine direction) × 0.81 m in CD (cross direction) × 0.15 m and five integrated shortwave infrared emitter twin-tubes (arranged across the conveyor belt with each (adjustable) 1.0 kW maximum heat output; effective emitting area: 0.50 m × 0.60 m). It was positioned at a distance of 150 mm over the belt and 200 mm after the fiber deposition point. By this means, the fibers were kept longer above the glass-transition temperature and, thus, had more time for the crystallization process. Thus, the heat shrinkage was reduced significantly. This has been demonstrated for aromatic polymers, as exemplified for PET, where heat shrinkage could be reduced from 50% (untreated) to 1–3% (with IR treatment), as well as for polyphenylene sulfide (PPS) and polyether ether ketone (PEEK), where it even could be reduced from >80% to around 1% in both MD and CD [26].

#### 2.3.3. Electrostatic Charging

The electrostatic charging of the nonwovens was carried out on a custom-made tailored set-up in analogy to a common industrial structure (Figure 3).

First, one side of the fabric (facing away from the test stand at the beginning) was positively charged by passing between the roller and the respective electrode (distance roll to electrode = “X”). Then, the counter side of the fabric was charged negatively by passing the next roller electrode pair. This created a stable charge separation (electret) in the flat material due to quasi-permanently aligned electric dipoles. This treatment positively influences the separation performance of the treated medium, especially with regard to charged particles, by means of a quasi-permanent electric field.

A “KNH 340” high-voltage generator (max. 30 kV, DC) of Eltex Elektrostatik GmbH (Weil am Rhein, Germany) was used to generate the required high voltage and applied to the fabrics by two electrodes “R131A” with a width of 0.50 m, also (Eltex Elektrostatik GmbH; Weil am Rhein, Germany), were connected to it. The strength of the applied electric field is determined by the distance “X” (Figure 3) between the electrodes and the earthed rollers and the set high voltage. To increase the electret effect, the conveying speed of the material was also reduced to a minimum (0.6 m min^–1^). Each fabric was loaded with an electrode voltage of +30 kV and accordingly, −30 kV (potential difference = 60 kV) using a roller electrode distance of 35 mm.

### 2.4. Testing of Physical and Mechanical Properties of Produced Nonwovens

#### 2.4.1. Fiber Diameter

The fiber diameter distribution was determined using scanning electron microscopy (SEM). A round sample was punched out of the nonwoven and placed on the SEM carrier, sputtered in an argon plasma (40 s under a vacuum of 0.1 mbar, with a distance of 35 mm, a current of 33 mA, and a voltage of 280 V) using a “Union SCD 040” of Balzers Union Ltd. (Balzers, Liechtenstein) with a gold–palladium layer of 10–15 nm. Three SEM micrographs per sample were taken with a magnification of ×1000 using a “TM–1000 tabletop electron microscope” of Hitachi High-Tech Corporation (Tokyo, Japan) with an accelerating voltage of 15 kV in the “charge-up reduction mode”. The magnification was chosen to catch around 40 single fibers per image. Contrast and brightness were adjusted to gain an image of straight monochromic fibers in front of a dark monochrome background. To analyze the images with regards to automated fiber diameter distribution, the beta software “MAVIfiber2d” of Fraunhofer ITWM (Kaiserslautern, Germany) was applied [39]. First, the images were smoothed by an algorithm and binarized by the software before statistical analysis was performed over each fiber pixel without segmentation into individual fibers [40,41]. After merging the output of the three images, the mean and median fiber diameter as well as the standard deviation and interquartile range were determined.

#### 2.4.2. Fabric Area Base Weight

The area base weight of nonwovens was determined referring to DIN EN ISO29073–1, adjusted by cutting out and weighing square sections of 100 cm^2^ (10 cm × 10 cm). To consider uniformity scattering along the cross direction (CD) of the nonwovens, three samples were taken in CD and averaged.

#### 2.4.3. Nonwoven Thickness

The thickness of the nonwoven fabrics was measured on the samples of the base weight measurements using a test head (Frank-PTI GmbH, Birkenau, Germany) of 25 cm^2^ and a test force of 5 cN cm^–2^. Eight measurements were executed diagonally along the sample, determining a median value for thickness (*δ*).

#### 2.4.4. Air Permeability

Air permeability was measured on the 10 cm × 10 cm sections in accordance with EN ISO 9237: 1995-12 with a test area of 20 cm^2^ and a differential pressure of 200 Pa.

#### 2.4.5. Filtration Efficiency

The mechanical filtration performance of the fabrics was examined using a filter media test system “HFP 2000”, equipped with an aerosol generator “PLG 2010” for the defined atomization of oils to disperse aerosol with a known and constant droplet diameter distribution and a “welas^®^ digital 2000” high-resolution aerosol spectrometers for aerosol/particle counting, aerosol/particle sizing, and research-grade aerosol/particle measurements (all PALAS GmbH, Karlsruhe, Germany). Measurements were carried out referring to ISO 16,890 (replacing EN 779/ASHRAE 52.2 (room air filter)), ISO 29,463 (replacing EN 1822–3 (HEPA-Filter)), and CEN EN 143:2000. In the context of the standard for filtering half mask classification (FFP according to EN 149), the test parameters were adapted to a test area of 100 cm^2^ (flat, circular rondes), volume flow of 95 l min^–1^, and a flow viscosity of 9.3 cm s^–1^ using liquid paraffine oil (density 0.825–0.850 g cm^–3^ at room temperature). It should be noted that the results cannot always be transferred exactly to ready-assembled masks, as parameters can change due to production conditions (e.g., due to stretching when joining a multi-layer media).

The fractional filtration efficiency was determined in accordance with EN 149 (EN:149: Section 8.11 with reference to EN 13274-7) after a 3 min lead time as an average over a measurement period of 30 s for a particle size range from 0.2 µm to 4 µm. The fractional filtration efficiency (*FR*) and pressure drop across the filter ∆*p* (measured before and after filter insert) were calculated from the raw- and clean gas concentration (see Equation (1)) concerning the raw gas concentration (*c*_0_) and the clean gas concentration (*c*_1_) as a function of the aerosol/particle diameter (*d_p_*) (Equation (3)), considering the results obtained with an empty container (i.e., without a filter sample).(3)$$FR(d_{p})\;=\frac{c_{0}\left(d_{p}\right)-c_{1}\left(d_{p}\right)\;}{c_{0}\left(d_{p}\right)}$$

The evaluation of the investigations includes *FE*_0.4µm__,_ the filtration efficiency up to the particle size 0.4 µm (≤0.4 µm corresponds to the MPPS (most penetrating particle size)), and *FE_tot_*, the total filtration efficiency over the particle size range 0.2–4.0 µm (see Equation (4)).(4)$${FR}_{tot}=\int_{0.2\;µm}^{4.0\;µm}FR\;(d_{p})\;∆d_{p}\Delta \;d_{p}$$

#### 2.4.6. Tensile Testing

Tensile tests of the nonwovens were carried out on an “Instron UPM 4301” of Instron GmbH (Darmstadt, Germany) to determine the tensile strength (*σ_m_*), the elongation at peak force (*ε_m_*), elongation at break (*ε_B_*), and the elastic modulus (*E*) as secant modulus (all in MD and CD). For each sample, five specimens with a width of 15 mm were cut out and tested in MD and CD, respectively. The sample thickness was determined individually according to DIN EN ISO 9073-2, and the median of five measurements was used for the calculation of the stress from the recorded force. Tests were executed with 100 mm min^–1^ using a 5 kN measuring head with pneumatic clamps (100 mm clamping length). The tenacity was calculated over the sample dimension, the fabric thickness, and the measured peak force. The median and the standard deviation of all measurements’ properties were used to compare the nonwoven characteristics.

#### 2.4.7. Thermal Stability

The heat shrinkage of the nonwovens was measured according to the “drying oven method” (ISO 11501:1995 and GB/T 12027–2004) [42] to determine the (time-)dimensional stability of the fabrics under thermal influence. Rectangular samples (300 mm × 50 mm in MD) were punched out on the left, in the middle, and on the right side of a nonwoven sample. The specimens were placed free-hanging in an oven at 200 °C. After 15 min, the samples were taken out, and the sample size was measured. The ratio of the dimensional change value to the size before shrinkage was calculated as the percentual shrinkage rate of the sample.

## 3. Results

### 3.1. PEF Synthesis

Two batches of PEF were synthesized. One batch was subjected to a subsequent SSP, while the other was used as received to give a PEF sample with high molar mass and a PEF sample with moderate molar mass, respectively. The characterization results of both charges are shown in Table 1 in comparison to the PET reference material.

Only PEF-1 data are shown in all columns as PEF-2 was not exposed to SSP in order to obtain one PEF with SSP and one PEF without SSP for the trials. For the commercial PET, the [η] and CEg specifications indicate no exposure to SSP and are presented in the respective column for comparison to the PEFs. Both PEF-1 and PEF-2 were amorphous and amber prior to crystallization. After the crystallization and SSP, the samples turned turbid and off-white.

Figure 4 shows the DSC curves of (first run of heat flux vs. temperature) PEF-1 after SSP (PEF-1_SSP_), PEF-2, and the commercial PET- and PBT reference materials.

The corresponding thermal properties and crystallinities are given in Table 2, and the X-Ray diffraction image and the characteristic X-Ray pattern of the PEF samples are shown in Figure 5.

Finally, the viscosity curves (absolute value of complex shear-viscosity) of all samples were determined as function of the temperature (Figure 6a). From these curves, the meltblow processing temperatures were derived. Furthermore, the thermal stability at a given process temperature was monitored over time in order to prevent degradation during the spinning process (Figure 6b).

The related plots of the storage and loss modulus are shown in Figure 7.

### 3.2. Meltblow Processing

The two different PEF charges, i.e., higher molar mass PEF-1_SSP_ and medium molar mass PEF-2, were processed on the 500 mm meltblow line equipped with a 28.4 hpi spinneret. The temperature profile of the extruder was set according to the rheological characterization (see Section 2.1) to obtain a melt temperature (*T_melt_)* of 270 °C. The die temperature (heat zones 6–8) was set around 5 K higher than the melt temperature. A heating ramp was applied along the extruder from the feed zone (zone 1: T = 245 °C) along the three extruder zones (zone 2: T = 250 °C to zone 5: = zone 6: T = 275 °C) to melt and homogenize the materials and reduce the thermal decomposition. Additionally, the process air temperature (*T_air_*) was fixed at around 15 K above the melt temperature, which was found to be the lowest temperature to still deliver a homogeneous air/melt flow out of the spinning beam. The temperature settings for all processed polymer samples are given in Table 3.

The PET and PBT reference materials were processed at previously developed [26] process settings.

Both PEF types showed a stable process with a stable and homogeneous flow of the melt/air stream from the die. The end gap was adapted in order to avoid melt adhesion on the air blades. PEF-1_SSP_ was processed with a 2.0 mm end gap, which was lower than the end gap of 3.0 mm employed for PET. The decreased end gap allows us to achieve lower fiber diameters as the air acts more efficiently to stretch the fibers. For PEF-2, an end gap as low as 1.5 mm was suitable to run a homogeneous process, equivalent to PBT (end gap = 1.5 mm).

The detailed process settings of all polymers are given in Table 4.

For PEF-1_SSP_, the applicable polymer throughput was limited to 0.023 g ho^–1^ min^–1^ (equals 0.4 kg h^–1^ m^–1^) as the defined process pressure limit of the spinneret (40 bar) was exceeded at higher values. PEF-2 and the reference samples showed no similar limitations due to their lower viscosities. The *DCD* was varied accordingly to optimize the web formation. For the same purpose, the infrared radiation power (Section 2.3.2) was adapted to optimize the web strength and fabric haptics (qualitatively). For PEF-2, strong turbulences (fly) in the deposition occurred, caused by process air overflow from the conveyor belt. As a result, the *DCD* was lowered from 150 mm to finally 80 mm. However, the low DCD prevented the usage of secondary air heaters due to the needed construction height. Nevertheless, tracking of the surface temperature of the nonwovens (using an infrared thermometer) along the conveyor showed a constant surface temperature of >135 °C at all points until exiting the “IR zone” (Figure 2).

### 3.3. Nonwoven Characterization

The characterization results of the nonwoven fabrics are summarized in Table 5.

Representative SEM micrographs of all settings are shown in Figure 8.

Of note, the sample MB-PEF1_SSP_-1 could be laid down and wound on the collector roll, but the fabric appeared as a very fluffy bulk of loose fibers with virtually no web strength. Thus, the sample was unusable for further processing and even for most of the characterization tests. Table 6 provides the mechanical properties of the nonwovens determined by tensile testing.

Additionally, WAXS and DSC measurements were performed on the PEF samples to reveal information about the degree of crystallinity and the lattice structure after fiber formation. The resulting measurement curves are shown in Figure 9.

The DSC heating thermograms of MB-PEF1_SSP_-1, MB-PEF1_SSP_-2, and MB-PEF2-1 show no pronounced melt peaks. However, in the range from 190 °C to 210 °C, diffuse peaks with enthalpies between 0.8 J g^–1^ and 1.9 J g^–1^ were observed, indicating the beginning of a crystallization process (small crystals). The WAXS diffraction images of MB-PEF1_SSP_-1, MB-PEF1_SSP_-2, and MB-PEF2-1 revealed an amorphous structure, similar to the one of MB-PEF2-2 (Figure 7a). Indeed, a melting peak of 9.3 J g^–1^ was observed in the DSC heating curve of MB-PEF2-2 with a peak temperature at 206 °C, which is 10 K lower than for the granule (compare Table 2), indicating a crystallinity of the fibers of 7%. For PEF2-2, the *T_g_* was equal for the fabrics, as well as for the respective granules but not higher than the *T_g_* of PET, which increased compared to the raw material (prior to processing). MB-PEF1_SSP_-1, MB-PEF1_SSP_-2, and MB-PEF2-1 even showed a lower *T_g_*. The PBT fibers showed a more distinct melt peak at 225 °C (equal to the raw material) with an enthalpy of 52.6 J g^–1^ (*Χ_c_ =* 37.6%). Also, the PET fabric showed a melting peak in the same temperature range as the granule (255 °C). The measured enthalpy of the peak was 38.3 J g^–1^, corresponding to a crystallinity of 27.4%.

### 3.4. Fabric Charging

During the charging of the fabrics, a current flow was present at the electrodes, when voltage was applied. It was 0.4 mA to 0.7 mA at the positively polarized electrode and 1.0 mA to 1.2 mA at the negative electrode indicating a successful charge carrier transfer to the medium. Table 7 shows the filtration efficiency (“total” and for singular for the 0.4 µm particle-size fraction) and the pressure drops for the status before and after the charge treatment.

Reducing the electrode distance led to hole formation due to a too excessive impact. In accordance with previous findings, increasing the number of passes mainly impacted the permanence of the electret effect, not its strength. The influence of the thickness is here negligible for the order of magnitude of the thickness of the meltblown fabrics. Plots of the filtration efficiency vs. the particle size fraction for the different nonwovens before and after electret treatment are shown in Figure 10.

## 4. Discussion

### 4.1. Material Characterization

The PEF synthesis on a relevant scale of 25 kg per batch was successfully carried out via the FDCA route. The resulting [η]-values of 0.53 dL g^–1^ and 0.56 dL g^–1^ are in line with the PET reference type, which was previously established as a meltblown grade. As already found earlier, the CEG content is lower in PEF obtained via transesterification of FDCA [12]. However, the intrinsic viscosity after SSP indicated high molar mass PEF with an [η] of 0.76 dL g^–1^. In line with the increase in molar mass (quantified by the viscosity in solution), the carboxylic acid end group content (CEG) decreased slightly from 29 µmol g^–1^ to 20 µmol g^–1^. Prior to SSP, both PEF charges were amber and transparent; after SSP, crystallin and off-white granulates were found.

The DSC curves of the PEF-2 granule showed no exotherms or endotherms in the first heating ramp, but a *T_g_*-step in the same range as PEF-1 after SSP (~86 °C). After crystallization (see a second heating ramp in Figure 4b), a broad melting peak with low intensity can be obtained as adjusted by the recrystallization peak before. This behavior emphasizes the slow crystallization of PEF. Similar to PET and PBT, distinct melting peaks were observed in the heat curve after SSP. As expected, the *T_g_* of PEF was higher compared to PET and PBT, combined with a shift of crystallite melting to a lower temperature (216 °C). WAXS measurements were consistent with previous findings, showing an amorphous signal after synthesis and distinct diffraction patterns of crystallized- and SSP-treated material. The crystallinity determined by WAXS (47%) was in line with the DSC-measured value of 45.2%. The crystalline reflections were observed at *2θ* = 16.2° ((101)), 17.9° ((004)), 19.4° ((11̲0)), 20.6° ((103)), 23.5° ((110)), and 26.9° ((020)) with a reflex at 19.4° as a typical marker, absent in the *α*′- and *β*-phase. However, as the expression of this reflex was very low for PEF-01_SSP_, e.g., compared to FDME-derived PEF [1], a mixture of *α*- and *α*′-phase is possible. The presence of a β-phase can be excluded as no reflex at *2θ* = 9.5° was observed. For the crystallized “non-SSP”-PEF-2, however, the reflex at 19.4° is very prominent indicating a more uniform or more distinct crystallization form of the shorter PEF chains.

The viscosity curves (Figure 6) confirmed the increase in molar mass. PEF-1_SSP_, which has undergone an SSP shows typical viscoelastic behavior of a thermoplastic melt (G’’ > G’,) over the full temperature range (compare Figure 7a) with an indication for starting degradation exceeding 310 °C (signal begins to “noise”/values begin to fluctuate). PEF-2, which was not supplied to an post-synthesis SSP step, shows a plateau of G’’ at around 250 °C, an increase in G’ from 240 °C to 275 °C, and a gelation point (G’ exceeding G’’) at 272 °C, which means that the melt has transitioned from fluid flow like behavior to solid elastic behavior. As the polymer still shows a high end-group concentration (see CEG in Table 1), this can be referred to as an ongoing molecular build-up of the melt under the prevailing measurement conditions, especially the low shear, but also states a limit for the processing. The PET reference material also shows typical viscoelastic behavior with a tendency for build-up (see Figure 7b) from 290 °C to 305 °C as sufficient concentration end-groups are present [34] and a starting decomposition > 320 °C, typical for PET. The commercial PBT shows a plateau for G’’ up to 275 °C, dominating the rheological behavior (see viscosity in Figure 6a). G’ shows a build-up of molar mass between 250 °C and 275 °C with a minor effect on the polymer’s viscosity as this effect lies two orders of magnitude below the level of the storage modulus. Above 300 °C, the material shows thermal degradation.

Derived from the upper viscosity limit for the meltblow process [28,33,43], the process temperatures were specified as >270 °C for PEF-1_SSP_ and >225 °C for PEF-2. The previously determined PET and PBT process temperatures of 255 °C and 280 °C (Table 3) were in accordance with viscosity measurements. Remarkable is the high sensitivity of the viscosity of PEF to temperature changes. This can be attributed to the lack of additives in “home-made” PEF compared to the commercial polyesters including additives. Although PEF-1_SSP_ and PEF-02 showed no critical time-dependent degradation over the first 15 min (typical retention times in the extruder), PEF-2 showed a higher viscosity in the time-sweep (by one order of magnitude, measured at 245 °C) compared to the temperature sweep. This indicates a time-/temperature-dependent degradation effect superimposing the temperature sweep. This in turn is counterbalanced by the onset of gel formation (see the increase in G’ from ~245 °C, Figure 7a) leading to the formation of the viscosity shoulder in the temperature sweep, which is also typical for PBTs as can be seen from the corresponding viscosity and G’ plots (Figure 6 and Figure 7).

### 4.2. Nonwoven Process and Fabric Characterization

PEF-1_SSP_ was processable at 272 °C, matching the viscosity window determined by the rheological characterization. However, the process productivity (maximal throughput) was limited to 0.023 g ho^–1^ min^–1^ by reaching the critical pressure value of 40 bar and thus stayed far below industrial standard values. This is due to the high intrinsic viscosity of 0.76 dL g^–1^ after SSP, which is already in the range of a melt spinning grade (compare [12]). The reference PET, selected as meltblow grade, showed an intrinsic viscosity [η] of 0.55 dL g^–1^ resulting in a process pressure of 12.0 bar at a > 4-times higher per-hole throughput leaving the possibility for further throughput increase by a multiple factor (at least by a factor of 5, which is the machine-specific limit).

In this context, the [η] of PEF-02 (0.56 dL g^−1^) was closer to that of the reference polyester grade (0.55 dL g^−1^). Accordingly, the process pressure of 23.4 bar at 0.1 g ho^–1^ min^–1^ for PEF-2 (equal to that of PET) at the same temperature setting as PEF-01_SSP_ showed potential for further throughput increase. However, the pressure was twice that of the commercial meltblow polymer, and the process temperature of 268 °C was higher than expected based on viscosity data. As discussed before, the temperature sweep was superimposed by thermal degradation and gel formation during the measurement. Higher process temperatures were avoided, as the gel point was detected at 273 °C in the temperature sweep (Figure 7).

Regarding the energy aspect of the processes for PEF and PET, PEF could be processed with the same productivity at a lower process temperature of 10 K and a lower air temperature of 15 K. When comparing PEF to PBT, the energy consumption was nearly counterbalanced by a 15 K higher melt, but 15 K colder process air.

A major difference between the PEF nonwovens can be seen in general in the resulting fiber distributions between the PEFs of different [η] (PEF1_SSP_ and PEF-2, Figure 8), which reveal the presence of both fine and coarse fiber diameters for MB-PEF1_SSP_-1 (Figure 8a) and MB-PEF1_SSP_-2 (Figure 8c). A further relevant difference can be taken from the processing setting of sample MB-PEF-1_SSP_v-1. Using a DCD of 500 mm marking the upper end of typical industrially applied settings, the fiber deposition results in a very loose, fluffy, and bulky fabric with almost no strength and handiness. This is consistent with the state of literature, reviewed by Kara and Molnar in 2022 [44] and stems from a reduced adhesion between the fibers as a result of lower fiber contact temperatures [45] as well as reduced fiber-entanglement [46,47] and bonding [48] as the fibers are deposited with lower drag force per air and less effect of the air pressure [49]. Furthermore, the air turbulence is reported to increase with higher DCDs lowering the packing density, increasing the thickness [50], and lowering the web strength [51]. Related to that, an increasing DCD causes an increase in the average pore size [47,50,51] and also a broader deposition (in CD) [47,50]. The influence of the DCD on the fiber diameter is controversially discussed in the literature. While most authors report decreasing fiber diameters with higher DCD—within their examined range of variation (e.g., [50,51,52,53,54,55,56,57])—no influence of DCD on fiber diameters was reported too [48,49], and even an increasing effect [58] has been published. However, it has to be considered that the literature is related to different polymer systems. Indeed, Chen et al. [59] specified the effect of an increasing DCD to reduce the fiber diameter to end 140 mm below the spinneret. Surprisingly, the results of Bo et al. [60] showed, for very high DCDs, that the fiber diameters increase again above 1000 mm due to the effects of turbulence after they decrease linearly between 700 mm and 1000 mm. Furthermore, Bresee and Qureshi [56] only found the coefficient of variation of the fiber diameter to increase with DCD due to fiber fusion during flight. This is applicable to PEF as well. While the median fiber diameter is almost identical for MB-PEF1_SSP_-1 and MB-PEF1_SSP_-2, the mean average is higher for a higher DCD. Also, by further lowering the DCD to 80 mm with MB-PEF02-01, the median results are again lower. This may be mainly due to the lower polymer viscosity. However, the fact that the mean average also lies significantly closer to the median implicates a narrower fiber diameter distribution, which can also be seen by comparing the SEM images of the nonwovens made of PEF-01_SSP_ (Figure 8a–d) and of PEF-2 (Figure 8e–h), respectively, thus proving the findings of Bo et al. [60].

As expected, the lowest median fiber diameter and most homogeneous fiber deposition (Figure 8g) were observed for PEF-02 with an increased air volume flow (maximal setting available at the machine) achieving 2.04 µm as median and 2.42 µm as mean. The median fiber diameters were in the same range as observed for the PBT reference material. Using PET coarser fibers were laid down, consistent with the state of the art [44,61]. However, the air permeability of all nonwovens was comparable. The lowest value was observed for PBT (729 l m^–2^ h^–1^), indicating the highest homogeneity, as also depicted in the CV of the base weight. MB-PEF2-2 shows a lower air permeability than PET (factor 1.3) despite the slightly, but significantly higher base weight value of PET (factor 1.2). This may be due to the, on average, lower fiber diameters and thus higher pore volume size.

For the reference polyesters, the heat shrinkage tests proved the suitability of the infrared heaters (in combination with the lower flight time by low DCD) to transfer sufficient heat to the PET fibers, thereby providing the required crystallization time. Accordingly, the faster crystallization rate of PBT also led to a higher degree of crystallinity (*Χ_c_ =* 37.6%, Figure 9d). MB-PEF1_SSP_-1 reacted with high shrinkage to the thermal exposure, because the infrared heaters could not be applied due to air turbulences over the conveyor belt, causing fluffy fiber material to cool below *T_g_* already at the deposition point. In general, the PEF fabrics remained far behind the shrinkage results obtained with PET and PBT. MB-PEF2-2 showed an improved heat resistance (lower shrinkage) correlating with an increased degree of crystallinity (Figure 9c,d). The higher process air volume (and higher air pressure and speed) resulted in higher stretching and chain orientation of the fibers, which, on the one hand, slightly reduced the fiber diameters (Table 5) while increasing the mechanical strength. On the other hand, reducing the applied temperature drop of the secondary air is supposed to cause a delay in strain-induced crystallization and, thus, insufficient crystallinity to suppress the shrinkage further [62]. According to Rieger [26], a minimum degree of crystallinity of 25% is needed to eliminate shrinkage. MB-PEF02-02, however, only showed crystallinity of 7% in DSC revealing insufficient crystallization progress. The corresponding WAXS diffractogram showed an amorphous signal. This is consistent with low and partially oriented PEF yarns [1,62], which also displayed diffuse endothermal crystallite melting peaks in DSC, but seemed to be entirely amorphous in X-Ray diffraction. In contrast, reference polyesters had distinct crystallite melting peaks, with shrinkage values of 2% (MD and CD) for PET and 5% for PBT.

The discrepancies between DSC and WAXS signals for highly stretched PEF (e.g., MB–PEF2–2) are attributable to its slow crystallization. Consequently, the fibers are characterized by highly oriented polymer chains in the fiber axis direction (amorphous orientation) and a low amount of probably small, stress-/strain-induced crystals. These crystals are insufficient to interact with the X-Ray signal but consume energy during DSC melting. A similar XRD limitation to determine low crystallinities below 10% has been previously reported [63]. Here, it was shown that the weak crystalline reflections are submerged in scattering from the amorphous fraction.

Nevertheless, it has to be noted, in the current state of our research, that the applied temperature management offered less time to obtain sufficient crystallized PEF fabrics. In this context, the temperature significantly impacts crystallization, whereas isothermal residence time has a comparatively minor effect. Exemplarily, Rieger [26] obtained a doubled degree of crystallinity by increasing the crystallization temperature of PET from 120 °C to 140 °C, while doubling the residence time only resulted in a 50% increase, from the same starting value. Ideally, the deposition temperature should align with the temperature of maximum crystallization speed, which is for PET between 170 °C and 190 °C and has to be maintained until winding. Rieger further observed that at 180 °C with an isothermal time of 40 s, 12 g m^−2^ PET fabrics achieved sufficient degrees of crystallinity (~35%) and eliminated heat shrinkage. For 40 g m^−2^, equivalent crystallization under the same conditions required 108 s of isothermal time, without degradation in either case.

While PBT exhibited fine fibers and the highest crystallinity, its mechanical properties, particularly in MD, were poor. Although PBT formed homogeneously deposited fabrics, rapid cooling and crystallization resulted in loosely entangled, fluffy fine fibers with limited tensile strength. This limitation is less critical for applications such as inner layers of multi-layer constructions or could be mitigated by post-processing, like calendering. In contrast, MB-PEF2-2 competed with the PET reference, demonstrating superior tenacity, modulus, and elongation in both MD and CD directions (Table 6). The MD–CD tenacity ratio (~1:3) is typical for meltblown fabrics but could benefit from optimization, such as varying DCDs or process air temperature. The advantage of MB-PEF2-2 among the other PEF samples can be attributed to its denser packing and higher fiber stretching.

A significant positive influence of electrostatic charging was observed for all samples. MB-PEF2-2 showed superior filtration performance comparable to PBT and slightly improved filtration performance compared to PET; the other PEF fabrics fell behind. The superiority of MB-PEF2-2 can be explained by the smallest fiber diameter, the most homogeneous fiber deposition, and the smallest pore sizes. The same trend was observed after charging, increasing the total filtration efficiency to >90% for PET, PBT, and MB-PEF2-2. Similarly, MB-PEF2-1 and MB-PEF01_SSP_-02 showed an inhomogeneous fiber deposition with a mixture of fine and coarse fibers (Figure 8a,c). Hence, a stable and effective charge carrier transfer is possible for PEF, despite the stronger pronounced conjugated π-electron systems in PET and PBT compared to PEF.

## 5. Conclusions

In this work, we successfully demonstrated the meltblow processing of poly(ethylene furanoate). The study showed that PEF is competitive with PET in all aspects of the meltblow process:access to industrial relevant (per-hole) throughputsdelivering low fine (micro) fiber diameters (median and mean average)comparable mechanical strength in MD and CD and air permeability values to PETsignificant filtration performance before and after electret treatment.Indeed, advantages over both PET and PBT have been identified, such as:lower processing temperature (energy consumption) compared to PETsuperior mechanical performance compared to PBT at equal energy consumption.

One aspect of PEF that still needs to be improved is its heat shrinkage. The potential to overcome this limitation has already been suggested by using longer post-heating distances after fiber laydown due to the slower crystallization time. Furthermore, SSP is not required for successful meltblow processing of PEF. The state of the art so far has been in the generation of PEF for textile and technical yarns, where, at the moment, an SSP is absolutely necessary for successful processing.

We expect the results presented to push FDCA and PEF production to larger scales. This step will be absolutely necessary for PEF to enter the nonwoven sector, as single meltblow plants consume several 100 kg up to tons per day.

## Figures and Tables

**Figure 1 materials-18-00544-f001:**
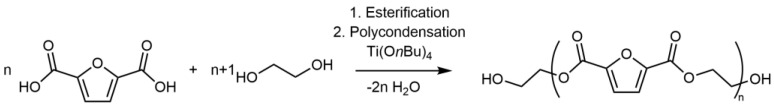
Synthesis of PEF from FDCA route.

**Figure 2 materials-18-00544-f002:**
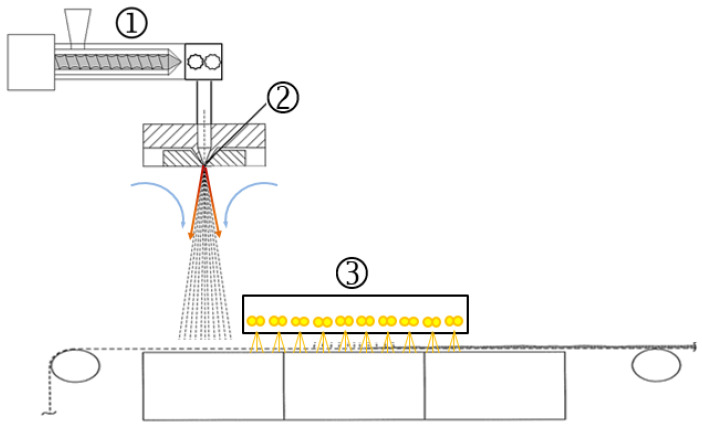
Schematic illustration of the meltblow system with additional secondary air heating and IR post-treatment.

**Figure 3 materials-18-00544-f003:**
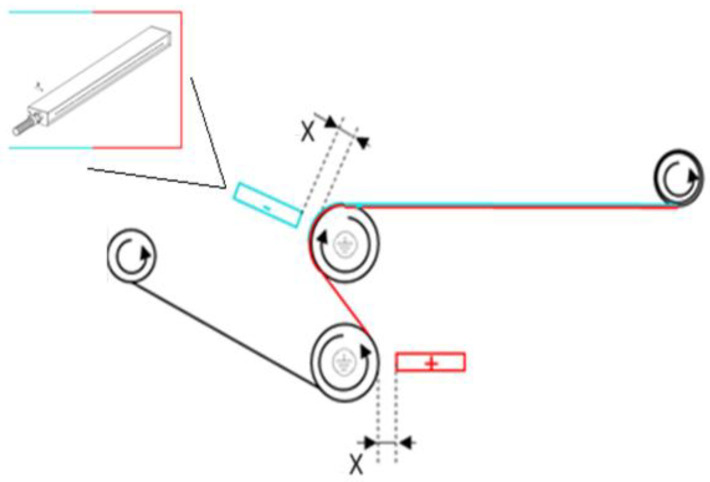
Illustration of the electret post-treatment for nonwoven charging.

**Figure 4 materials-18-00544-f004:**
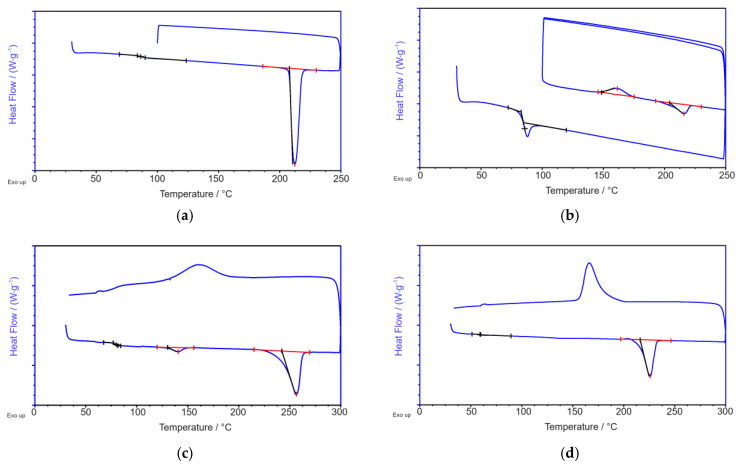
DSC curves of (**a**) PEF-1_SSP_, (**b**) PEF-2, (**c**) PET (Advanite 64001), and (**d**) PBT (Pocan B600).

**Figure 5 materials-18-00544-f005:**
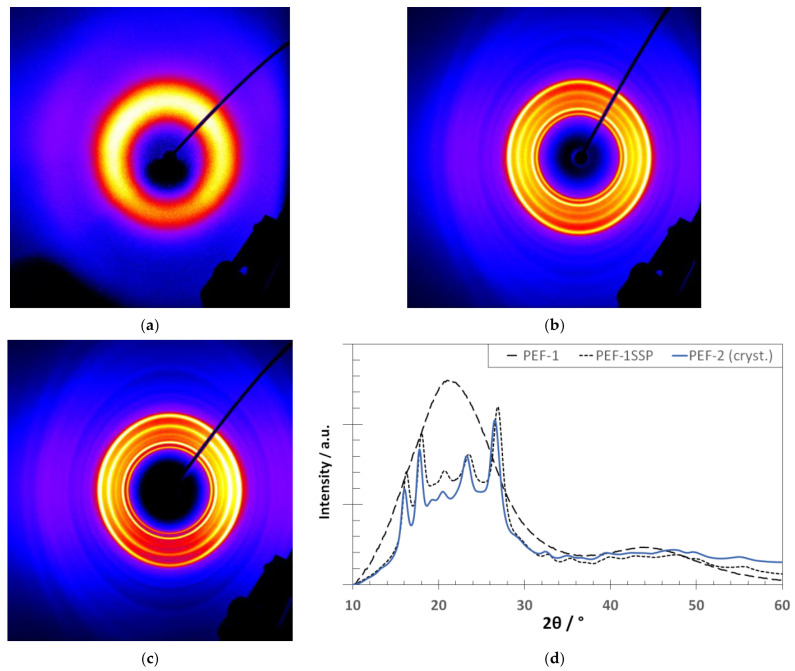
X-Ray images of the PEF samples: (**a**) diffraction image of PEF-1 as synthesized, (**b**) diffraction image of PEF1_SSP_, (**c**) diffraction image of PEF-2 (crystallized), and (**d**) corresponding X-Ray patterns.

**Figure 6 materials-18-00544-f006:**
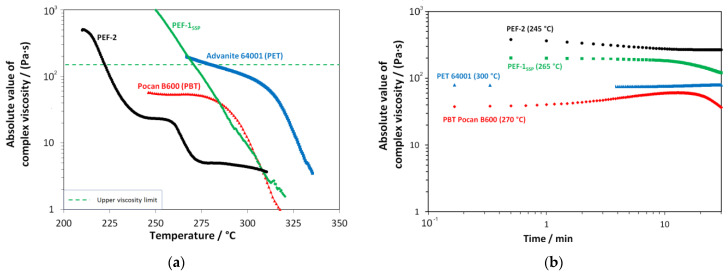
Shear rheological characterization of black PEF-11_SSP_, green PEF-2, blue PET reference, and red PBT reference: (**a**) temperature sweeps (ω = 10 rad s^−1^, ε = 10%, $\dot{T}\;$ = 2 K min^−1^) for the determination of the process temperature windows (dotted line: min. process temperature) and (**b**) time sweeps at relevant temperatures for processing (ε = 10%, ω = 10 rad s^−1^) for the estimation of thermal degradation behavior under shear.

**Figure 7 materials-18-00544-f007:**
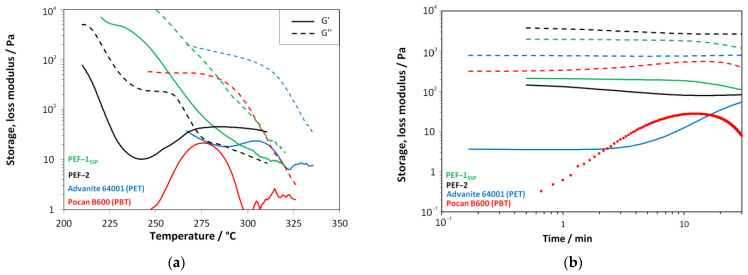
Plot of storage (G′) and loss modulus (G″) of the used polymers obtained from shear rheological characterization; black: PEF-1_SSP_; brown: PEF-2; blue: PET Advanite 64001; red: PBT Pocan B600: (**a**) temperature sweeps (ω = 10 rad s^–1^, ε = 10%, $\dot{T}\;$ = 2 K min^–1^) and (**b**) time sweeps at relevant temperatures for processing (ε = 10%, ω = 10 rad s^–1^).

**Figure 8 materials-18-00544-f008:**
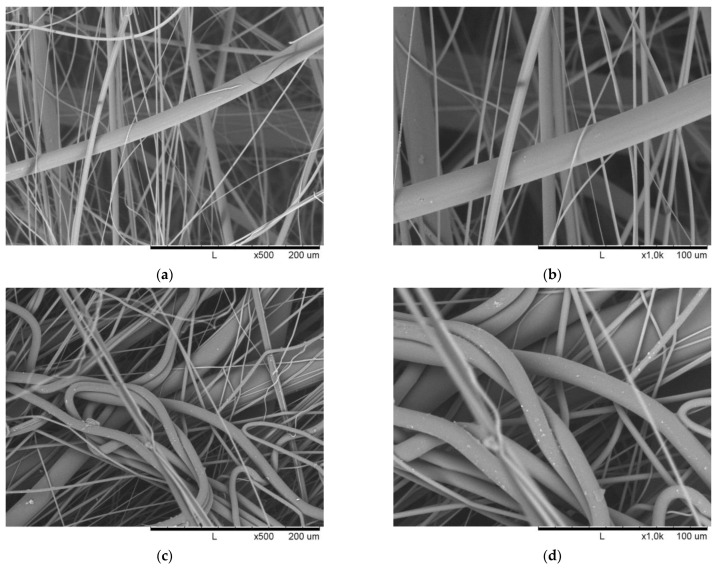
SEM micrographs of the produced nonwoven samples: (**a**) MB-PEF1_SSP_-01 (×500), (**b**) MB-PEF1_SSP_-01 (×1000), (**c**) MB-PEF1_SSP_-02 (×500), (**d**) MB-PEF1_SSP_-02 (×1000), (**e**) MB-PEF2-01 (×500), (**f**) MB-PEF2-01 (×1000), (**g**) MB-PEF2-02 (×500), (**h**) MB-PEF2-02 (×1000), (**i**) MB-PET (×500), (**j**) MB-PET (×1000), (**k**) MB-PBT (×500), and (**l**) MB-PBT (×1000).

**Figure 9 materials-18-00544-f009:**
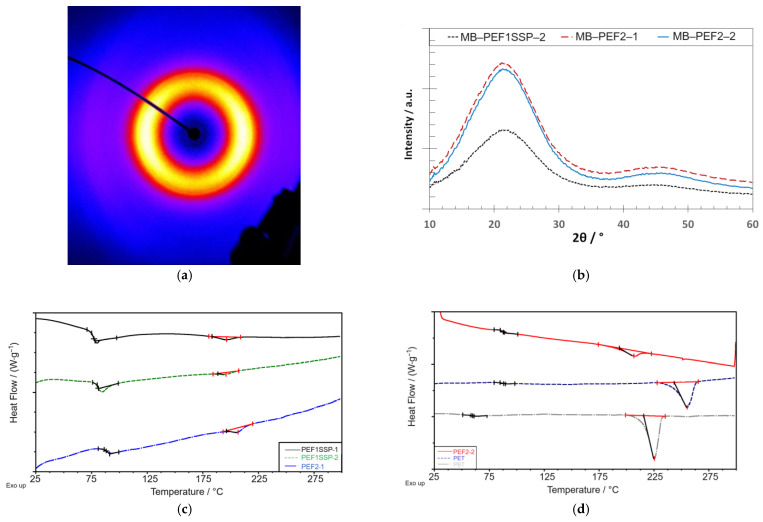
Structure analysis of the PEF nonwoven samples: (**a**) X-Ray diffraction image (exemplary for all measurements), (**b**) X-Ray patterns, (**c**) DSC thermograms of MB-PEF1SSP-1 (black), -2 (green), and MB-PEF2-1 (blue), and (**d**) DSC thermograms of MB-PEF2-2 (red) and PET-(blue) and PBT-(grey) reference fabrics.

**Figure 10 materials-18-00544-f010:**
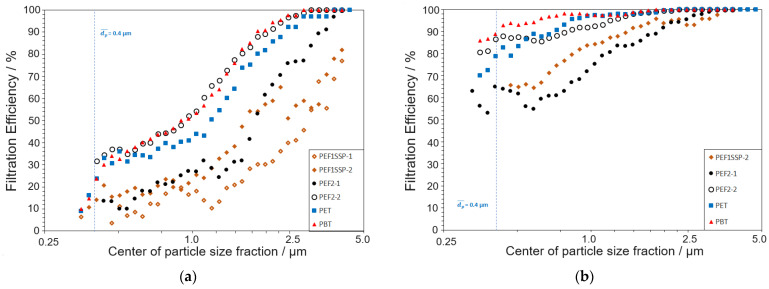
Plots of filtration efficiency vs. particle size fraction for the nonwoven samples: (**a**) before and (**b**) after electret treatment.

**Table 1 materials-18-00544-t001:** Summary of intrinsic viscosities [η], carboxyl end-group contents, (CEGs), molar mass number-averages *M_n_*, and dispersities Đ of PEF samples before and after SSP.

Sample	PEF-1	PEF-2	PET ^1^
[η] _(before SSP)_/(dL g^1^)	0.53	0.56	0.55
CEG _(before SSP)_/(µmol g^–1^)	29.5	38.5	30
[η] _(after SSP)_/(dL g^−1^)	0.76	-	-
CEG _(after SSP)_/(µmol g^−1^)	20.0	-	-
Mn _(before SSP)_/(g mol^–1^)	30,182	18,537	-
Đ _(before SSP)_	345	333	-
Mn _(after SSP)_/(g mol^–1^)	36,372	-	-
Đ _(after SSP)_	297	-	-

^1^ Reference material. Data from the supplier data sheet.

**Table 2 materials-18-00544-t002:** Thermal and structural properties of the synthesized PEF samples as determined by DSC; *T_g_*: glass transition temperature; *T_m,p_*: melting peak temperature; Δ*H_m_*: melt enthalpy; *χ_c_*: crystallinity.

Sample	*T_g_*/°C	*T_m,p_*/°C	Δ*H_m_*/(J g^–1^)	*Χc*/%
PEF-1_SSP_	86.4	212.3	61.9	45.2
PEF-2	85.7	215.7	40.7	29.7
PET ^1^	79.9	256.3	45.6	32.6
PBT ^1^	59.5	225.7	56.7	39.1

^1^ Commercial reference materials.

**Table 3 materials-18-00544-t003:** Processing temperatures of the processed polymers.

Polymer	Extrusion Temperature Profile/°C					*T_melt_*	*T_air_*
	Zone 1	2	3	4	Zone 5–8		
PEF-1_SSP_	245	250	255	260	275	272	285
PEF-2	245	250	255	260	275	268	285
PET	300	305	295	280	280	280	300
PBT	240	245	250	255	260	255	300

**Table 4 materials-18-00544-t004:** Process settings for the produced nonwoven samples.

Nonwoven Sample	End Gap/mm	Per-Hole-Throughput/(g ho^–1^ min^–1^)	Die Pressure/bar	DCD/mm	Air Volume Flow/(Nm^3^ h^–1^)	IR Treatment	
						Thermal Flux Density/(kW m^–2^)	Exposure Time ^1^/s
MB-PEF1_SSP_-1	2.0	0.023 ^2^	37.3	500	240	*Not applied*	
MB-PEF1_SSP_-2	2.0	0.023 ^2^	37.3	150	240	4.12	72
MB-PEF2-1	1.5	0.102	23.4	80	240	6.25 ^4^	27
MB-PEF2-2	1.5	0.102	23.4	80	325	6.25 ^4^	27
MB-PET	3.0	0.102	12.0	150	170	3.75	60
MB-PBT	1.5	0.097 ^3^	15.0	250	240	*Not applied*	

^1^ Time above *T_g_*: distance starting from fiber deposition point to after passing the “IR zone” = 0.9 m. ^2^ Maximal applicable throughput. ^3^ The deviation of the gravimetric throughput of PBT to PEF-2 and PET results from the specific density. ^4^ Higher IR power resulted in paper-like fabric haptic.

**Table 5 materials-18-00544-t005:** Nonwoven characteristics.

Sample	Base Weight/g m^2^		Thickness/mm	Fiber Diameter/µm		Air Permeability/(L m^–2^ h^–1^)	Heat Shrinkage (200 °C, 15′)/%
	Median	CV ^1^		Median	Mean		MD/CD
MB-PEF1_SSP_-1	24	10	59 ± 13 ^2^	2.79	4.61	– ^3^	– ^3^
MB-PEF1_SSP_-2	25	20	118 ± 28	2.93	4.20	2300 ± 900	42 ± 3/36 ± 13
MB-PEF2-1	27	15	118 ± 27	2.56	2.90	2010 ± 350	32 ± 9/44 ± 4
MB-PEF2-2	28	14	111 ± 16	2.04	2.42	1030 ± 122	23 ± 19/44 ± 3
MB-PET	33	12	97 ± 26	3.52	4.12	1320 ± 260	2 ± 1/2 ± 0.5
MB-PBT	27	8	125 ± 21	2.42	3.27	729 ± 90	8 ± 1/12 ± 3

^1^ Coefficient of variation (standard deviation divided by mean average). ^2^ Fabric is strongly compressed by the measurement force by a factor of 2 to 4. ^3^ Fabrics are too fragile and fluffy for further treatment due to insufficient web strength.

**Table 6 materials-18-00544-t006:** Mechanical properties of the nonwovens in MD and CD. *σ_m_*: tenacity; ε_peak_: elongation at max. force.

Sample	*σ_m_*/(N mm^–2^)		E-Modulus/(N mm^–2^)		ε_peak_/%	
	MD	CD	MD	CD	MD	CD
MB-PEF1_SSP_-1	– ^1^	– ^1^	– ^1^	– ^1^	– ^1^	– ^1^
MB-PEF1_SSP_-2	4.8 ± 2.0	0.4 ± 0.1	73 ± 58	15 ± 2	17 ± 6	43 ± 11
MB-PEF2-1	4.1 ± 0.4	1.8 ± 0.2	219 ± 30	71 ± 8	2 ± 1	10 ± 4
MB-PEF2-2	9.5 ± 1.0	2.9 ± 0.3	346 ± 48	89 ± 6	28 ± 5	48 ± 21
MB-PET	9.5 ± 0.8	3.0 ± 0.1	392 ± 50	88 ± 10	25 ± 14	41 ± 22
MB-PBT	0.5 ± 0.2	2.0 ± 0.2	85 ±32	15 ± 2	5 ± 2	82 ± 6

^1^ Fabrics that are too fragile and fluffy for further treatment due to insufficient web strength.

**Table 7 materials-18-00544-t007:** Filtration performance of the meltblown nonwovens before and after electret treatment.

Sample	*FE_tot_*/%		*FE*_0.4µm_/%		Δp/Pa	
	Uncharged	Charged	Uncharged	Charged	Uncharged	Charged
MB-PEF1_SSP_-1	15.6	– ^1^	5.5	– ^1^	2.4	– ^1^
MB-PEF1_SSP_-2	25.7	79.7	13.3	82.8	7.3	7.5
MB-PEF2-1	25.3	80.9	24.6	62.0	8.5	8.5
MB-PEF2-2	52.9	91.4	35.2	85.3	19.5	18.3
MB-PET	45.2	91.1	21.0	78.0	20.7	22.0
MB-PBT	52.0	97.1	25.0	86.1	23.1	23.2

^1^ Fabrics that are too fragile and fluffy for further treatment due to insufficient web strength.

## Data Availability

Data are available on request due to privacy restrictions. The data presented in this study are available on request from the corresponding author. The data are not publicly available due to running project issues.

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
