# Peer review of "Meltblow Processing of Poly (Ethylene Furanoate)–Bio-Based Polyester Nonwovens"

_materials, 2025, doi:10.3390/ma18030544_

Round 1
Reviewer 1 Report
Comments and Suggestions for Authors
Dear Author,
Your study investigate the processing of PEF nonwovens in the meltblow process. The resulting fabrics achieved median fiber diameters of 2.04 μm, comparable to PET. The filtration efficiency of 25 g m-2–fabrics exceeded 50 % - comparable to PET and PBT of same grammage - and was raised to over 90 % with post-process electrostatic charging, maintaining stability. However, the suppressed ring rotation and slower crystallization kinetics of PEF showed the need for longer post-treatment times as the heat shrinkage remained between 20 % and 40 %. at 10 °C. Overcoming this, PEF can be a viable, bio-based alternative to PET, particularly for such high-temperature nonwoven applications that require thin layers.
The methods are described in a good way. The effects are not fully discussed. The important figures are not inside. The conclusion is clear. The references are up to date. Overall, it is a paper with a lower quality, it necessary to make major corrections.
.
I have several remarks:
In the paper you have a reference to figure A1 in the Appendix A, you have section before the references and after Conclusion. Why? Please shift this section in the paper or delete it.
Table 1: I miss the explanation why you completely measure only one! Sample 1.
Table 2: You show only the data of PEF1ssp and PEF2 not PEF1 or PEF2ssp
Table 3: You show only the data of PEF1ssp and PEF2 not PEF1 or PEF2ssp
Table 4: No explanation below the table about the missing of data for some samples
Table 5: I get a value of -2 Why? Below the table you write twice explanations
2 Fabrics too fragile 435
and fluffy for further treatment due to insufficient web strength.
2 Coefficient of Variation (standard deviation divided by mean average)
Overall until table 4 you have fully characterized only sample 1ssp, and sample 2. I miss sample 1 and sample 2ssp. In the second part you have MB-Samples but extrusion you make only with sample 1 and sample 2ssp. Please add severall sentences about the experimental setup.
BR
The reviewer
Author Response
Dear Reviewer,
Thank you very much for the time and effort to review our manuscript and to point out improvements in our script. We apologize that some points in the manuscript have obviously remained unclear and we revised the paper accordingly to the comments in order to eliminate these points. Please find subsequent the point-to-point answers on your comments. I have several remarks:
Comment 1: In the paper you have a reference to figure A1 in the Appendix A, you have section before the references and after Conclusion. Why? Please shift this section in the paper or delete it.
Answer 1: Thank you for the comment. We shifted the section accordingly to your recommendation below the first reference on Figure A1 (now Figure 7) (l. 391). The respective discussion on the Figure is shifted to the discussion section, l.521f.
Comment 2: Table 1: I miss the explanation why you completely measure only one! Sample 1.
Answer 2: The explanation can be found in the materials section. PEF-1 is just a preliminary stage of PEF-1SSP, while PEF-2 was not exposed to a solid-state polycondensation (SSP) after synthesis in ordert o obtain one PEF with SSP and one PEF without SSP. PET was a commercial polymer, so we had no access to the synthesis, only to the final specification. As PET in this i.V. range has probalbly not been exposed to SSP, the values are presented at this column. We added a respective paragraph below the table to make this clear, „Only for PEF-1 data is shown in all columns as PEF-2 was not exposed to SSP in order to obtain one PEF with SSP and one PEF without SSP for the trials. For the commercial PET, the [η] and CEg specification indicates no exposure to SSP and is presented in the respective column for comparision to the PEFs.“ See l.356f
Comment 3: Table 2: You show only the data of PEF1ssp and PEF2 not PEF1 or PEF2ssp
Comment 4: Table 3: You show only the data of PEF1ssp and PEF2 not PEF1 or PEF2ssp
Comment 3/4: The reason for this is the same as for the prior comment. PEF-1 is a preliminary stage for PEF-1SSP, while there was no SSP of PEF-2 and thus no PEF-2SSP.
Comment 5: Table 4: No explanation below the table about the missing of data for some samples
Answer 5: There is no missing data. For the respective sample, no I treatment was used as. We revised the table by the comment “not applied” at the respective position to avoid
Comment 6: Table 5: I get a value of -2 Why? Below the table you write twice explanations
2 Fabrics too fragile 435
and fluffy for further treatment due to insufficient web strength.
2 Coefficient of Variation (standard deviation divided by mean average)
Answer 6: Thank you for he comment! We revised the table accordingly and appologize for the error and confusion.
Comment 7: Overall until table 4 you have fully characterized only sample 1ssp, and sample 2. I miss sample 1 and sample 2ssp. In the second part you have MB-Samples but extrusion you make only with sample 1 and sample 2ssp. Please add severall sentences about the experimental setup.
Answer 7: We guess, here you mixed up the polymers (PEF-1SSP and PEF-2) with the nonwoven samples (MB-PEF1SSP-1 , MB-PEF1SSP-2, MB-PEF2-1 , MB-PEF2-2) All meltlbown samples have been characterized. These are MB-PEF1SSP-1 and MB-PEF1SSP-2, which were processed from PEF-1SSP and MB-PEF-2-1 and MB-PEF-2-2 from PEF-2. Indeed, MB-PEF1SSP-1 was too fragile to be measured by the air permeability under the applied pressure, which is already amrked at the table (see superscript)
The experimental set-up is detailed given in the methods section and the sample production parameters for all samples are given in Table 4.

Reviewer 2 Report
Comments and Suggestions for Authors
I have read the manuscript provided by the authors, and I have to say that it is in a good direction in finding substitutes for PET and related materials. The authors have prepared a really lovely manuscript. However, some points need clarification. Please check the pdf attached with all the comments.

Author Response
Dear Reviewer
Thank you very much for the positive feedback. Please find subsequent the answers on your comments.
Comment 1: Title: Polyethylene and not polyethylene. Please correct it.
Answer 1: Thank you for the note. We guess you meant “Polyethylene and not polyethylene”. We apologize for the error, which we corrected respectively.
Comment 2: Page 2, lines 48-49: Why the authors use Kelvins here?
Answer 2: We use “Kelvin” as the SI base unit of thermodynamic temperature. Despite that, we use “°C” for the absolute temperature values to be easier understandable – because the °C-scala is here more established - for the readers and “K” for temperature differences/gradients. We also used both units accordingly in earlier publications.
Comment 3: Authors are using C and K for temperatures. Better to only choose one.
Answer 3: Thank you for the comment. See accordingly answer to comment 2. In our opinion it is understandable for a scientific audience to differ between °C and K.
Comment 4: Page 2, lines 66-68: Please add references.
Answer 4: Thank you for the comment. We added a reference to the paragraph: Statton, W. O. (1967). Coherence and deformation of lamellar crystals after annealing. Journal of Applied Physics, 38(11), 4149-4151.
Comment 5: MEG sued as abbreviation; please define it before sue.
Answer 5: MEG is already introduced in l.29: “ethandiol (monoethylene glycol, MEG)“
Comment 6: Authors use syntheses and synthesis. Please choose one!
Answer 6: “Syntheses” is the plural form of “synthesis”. In our opinion we use the terms right.
Comment 7: 1.2: if any protocol was used for the synthesis, please add references.
Answer 7: Thank you for the comment. No protocol from literature was used for the synthesis, which is based on own project work. The procedure is given in the methods section (2.1.2., l. 103f) and is based on our former publication (1. Höhnemann, T.; Steinmann, M.; Schindler, S.; Hoss, M.; König, S.; Ota, A.; Dauner, M.; Buchmeiser, M. R. Poly(ethylene furanoate) along its life-cycle from a polycondensation approach to high-performance yarn and its recyclate. Materials 2021, 14(4), 1044.), which is already cited. We added the citation additionally to the synthesis section: “The synthesis protocol is based on earlier publication [1].” (Line 104f)
.
Comment 8: Please define all abbreviations before use.
Answer 8: We defined all abbreviations at the position of their first use. We screened our manuscript and found no abbreviation, which was not introduced.
Comment 9: 2.2.6: If I understand correct the authors dissolved PEF to 80-120 Celsius but then the GPC was measured at 50 Celsius? Is this correct? Because it is a little problematic to use two different temperatures fir dissolving and characterizing in GPC. This type of characterization is used for the commercial products as well? Is PET dissolved and passing through GPC at 50 Celsius?
Answer 9: Thank you for the detailed comment and remark. Yes, you understood the procedure right. We need the higher temperatures to get the sample into solution and to be able to filter it. The reason for the have lower temperatures in the GPC is that the cresol degrades in the long term at such high temperatures. However, we observed no signs for incomplete dissolution or aggregation. We are now aware of the problematic of different temperatures for future measurements,
No, PET - as a commercial reference - was not dissolved nor measured by SEC in our study. The comparability to the PEF was established via the i.V. as specification from the supplier.
Comment 10: Table 1: Can authors provide the molecular weight of PET as well? Also, how did they measure the molecular weight of PEF?
Answer 10: Sincerely, the molar mass of PET can not be provided as it was used as commercial material. Only the intrinsic viscosity was provided by the supplier. The molar mass distribution of self-synthesized PEF, was determined by size exclusion chromatography as described in the methods section (2.2.6, l.187f)
Comment 11: Figure 4: Why the DSC graphs of PEF 1 & 2 look so different? In b Tm looks like having two different temperatures. Any comments?
Answer 11: Yes, of course we can comment these differences in detail: In the first heating ramp: PEF-1_SSP shows a melting peak, PEF-2 not. Both polymers don’t show recrystallization, when cooled down due to the slow crystallization kinetics, already known for PEF. When crystallized, the former amorphous PEF-2 shows a respective recrystallization (exothermal peak) followed by melting (endothermal peak) of the formed crystals. So we don’t observe two different melting temperature. We revised the respective discussion at l. 509f to make it more clear. Thank you!
Comment 12: In Table 5, the authors presenting some data, but they do not refer how they did find all the values. Moreover they present SEM images, but no discussion is presented. Please add some discussion to explain what can be seen.
Answer 12: All values were determined according to the methods, presented in detail in section 2.4 („Testing of Physical and Mechanical Properties of Produced Nonwovens“), l.277ff. Further, the Sem images are presented within the results, the discussion can be found in the discussion section, e.g. „A major difference between the PEF nonwovens can be seen in general in the resulting fiber distributions between the PEFs of different [η] (PEF1SSP and PEF-2, Figure 7), which reveal the presence of both, fine and coarse fiber diameters for MB–PEF1SSP–1 (Figure 7a) and MB–PEF1SSP–2 (Figure 7c).“ (l.583f), „However, the fact that the mean average also lies significantly closer to the median implicates a narrower fiber diameter distribution, which can also be seen by comparing the SEM images of the nonwovens made of PEF–01SSP (Figure 7a - d) and of PEF–2 (Figure 7e - h), respectively, thus proving the findings of Bo et al. [65].“ (l.611f), „As expected, the lowest median fiber diameter and most homogeneous fiber deposition (Figure 7g) was observed for PEF–02“ (l. 616f).
Comment 13: Table 5: In air permeability the first entry is -2. Is this correct? Because I believe it is the superscript. Then in Table there are two “2” as superscript. The PEF samples are gerenally more permeable from PET and PBT? Is this going to be a problem for packaging?
Answer 13: Thank you very much for this comment! You are right. Correct is “–“ for no measurement and “2” as superscript. We revised the table accordingly. We would not agree to your impression that PEF is generally more permeable to PET. When comparing MB-PEF2-2 to MB-PET, the PEF sample is less permeable. For packaging we see no issue as PEF indeed offers much higher gas barrier properties (several orders) to PET as you can find in various publications. However, to mix up inflow, flow around and diffusion through fibers/fiber networks should be avoided in this context.
Comment 14: Page 20, lines 609-611: Please add references.
Answer 14: Thank you for the comment! We added a reference to a related publication, which describes the stated correlations fundamentally (Kraus, G., & Gruver, J. T. (1972). Kinetics of strain‐induced crystallization of a trans‐polypentenamer. Journal of Polymer Science: Polymer Physics Edition, 10(10), 2009-2024.) on the respective paragraph.

Round 2
Reviewer 1 Report
Comments and Suggestions for Authors
Now it is okay